# Empathy training for service employees: A mixed-methods systematic review

**Mathieu Lajante[1]\*, Marzia Del Prete[2], Beatrice Sasseville[3], Geneviève Rouleau[4], Marie-Pierre Gagnon[5], Normand Pelletier[6]**

**1** The emoLab, Ted Rogers School of Management, Toronto Metropolitan University, Toronto, Ontario, Canada, **2** Department of Economic Sciences and Statistics, University of Salerno, Fisciano, Salerno, Italy, **3** School of Psychology, Université Laval, Québec, Québec, Canada, **4** Nursing Department, Université du Québec en Outaouais, Québec, Canada, **5** Faculty of Nursing Sciences, Université Laval, Québec, Québec, Canada, **6** Business & Economics Librarian, Université Laval, Quebec City, Canada

\* Mathieu.lajante@torontomu.ca

**Data Availability Statement:** This is a narrative systematic review and there is no quantitative data associated to this paper. All the supportive

## Abstract

Following the surge for empathy training in service literature and its increasing demand in service industries, this study systematically reviews empirical papers implementing and testing empathy training programs in various service domains. A mixed-methods systematic review was performed to identify and describe empathy training programs and discuss their effectiveness in service quality, service employees' well-being, and service users' satisfaction. Included papers met those eligibility criteria: qualitative, quantitative, or mixed-methods study; one training in empathy is identifiable; described training(s) developed for or tested with service employees dealing with service users. We searched health, business, education, and psychology databases, such as CINAHL, Medline ABI/Inform Global, Business Source Premier, PsycINFO, and ERIC. We used the Mixed-Method Assessment Tool to appraise the quality of included papers. A data-based convergent synthesis design allowed for the analysis of the data. A total of 44 studies published between 2009 to 2022 were included. The narrative presentation of findings was regrouped into these six dimensions of empathy training programs: 1) why, 2) who, 3) what, 4) how, 5) where, and 6) when and how much. Close to 50% of studies did not include a definition of empathy. Four main empathic competencies developed through the training programs were identified: communication, relationship building, emotional resilience, and counseling skills. Face-to-face and group-setting interventions are widespread. Our systematic review shows that the 44 papers identified come only from health services with a predominant population of physicians and nurses. However, we show that the four empathic skills identified could be trained and developed in other sectors, such as business. This is the first mixed-methods, multi-disciplinary systematic review of empathy training programs in service research. The review integrates insights from health services, identifies research limitations and gaps in existing empirical research, and outlines a research agenda for future research and implications for service research.

information (e.g., search strategy, PRISMA, MMAT scores) have been submitted with the paper.

**Funding:** ML No 430-2019-00321, 2019 Social Sciences and Humanities Research Council of Canada https://www.sshrc-crsh.gc.ca/home-accueil-eng.aspx The funders had no role in study design, data collection and analysis, decision to publish, or preparation of the manuscript.

**Competing interests:** The authors have declared that no competing interests exist.

## 1. Introduction

Empathy—our ability to share and understand others' affective and mental states [1]—is regularly praised for its positive role in service interactions. It is a sociobiological process that engages service employees (SEs) and service users (SUs) in a reciprocity-based service interaction [2], enhancing SUs' experience and supporting firms' performance [3]. SEs' empathy elicits higher SUs' perception of service quality and satisfaction [e.g., 4], and service experience [e.g., 5]. However, empathy is not only a developmental trait but also a professional requirement that needs to be developed through empathy training programs, especially when SEs are asked to perform emotional labor through multiple interactions with emotional SUs. In a recent survey, 84% of CEOs reported that empathy is a crucial soft skill in customer service settings and an essential factor in improving the customer service experience [6].

Over recent years, numerous companies have implemented empathy training programs for their SEs. When Starbucks faced a wave of global outrage and a call for a boycott following the arrest of two Black customers in one of its Philadelphia stores, CEO Kenneth Johnson addressed this issue using an empathy-oriented intervention strategy. He enrolled approximately 175,000 SEs across the United States in a four-hour training session to build empathy, compassion, and a welcoming environment for all customers [7]. Boots, a UK-based pharmacy/cosmetics retailer, invested in a unique training program for SEs that includes empathy training to understand customer needs better [8]. Bank of America also developed a dedicated empathy training series called "Life Stages" for SEs to develop various soft skills. The empathy training program helps SEs examine the customers' needs throughout different stages of their life and allows SEs to build empathy through activities such as role-play [9]

Surprisingly, despite regular calls for investigating empathy training in service [e.g., 3], we found no empirical studies developing, testing, and implementing empathy training for SEs. Conversely, many studies in health have already investigated the efficiency of empathy training programs on physicians' and nurses' well-being and patients' satisfaction [e.g., 10]. Against this backdrop, we conducted a mixed-methods systematic review to identify, synthesize, and discuss the existing empathy training programs in various service contexts (e.g., health, marketing, education, and management). The mixed-methods systematic review had two main objectives: 1) identify and synthesize the existing empathy training programs empirically tested service delivery, and 2) discuss the effectiveness of those empathy training programs in perceived service quality, perceived value, and user satisfaction. We also wanted to discuss the applicability of those identified empathy training programs to various service contexts, such as service marketing.

The study addresses two gaps in the service literature. First, our systematic review is the first to synthesize empathy training programs in various service sectors and to discuss the results from a service marketing perspective. Second, our systematic review goes beyond the call for empathy training for SEs and introduces clear directions for implementing empathy training programs in service and a research agenda for future investigations that will foster the development and successful implementation of empathy training programs in services marketing. The rest of the paper is structured as follows: section 2 describes the methodology of the mixed-methods systematic review. Section 3 presents the study results. We expose the methodological quality assessment of included studies, describe the empathy training programs through the Template for Intervention Description and Replication (TIDieR) checklist [11], and report their effectiveness. Section 4 discusses the results and outlines a research agenda to foster future investigations and implementations of empathy training programs in services marketing. Finally, section 5 presents the limitations of our study.

## 2. Methodology

### 2.1. Formulation of the research question

Our mixed-methods systematic review targets empathy training programs in service (nursing, medicine, health delivery, business, education) defined as any training method (e.g., didactic, experiential, or mindfulness) dedicated to fostering, developing, or improving SE's empathic skills and to providing clear directions and recommendations in training SEs in empathic skills [12]. We formulated the research questions from discussions between the authors: 1) What is the existing empathy training program empirically tested about service delivery? And 2) How effective are empathy training methods for perceived service quality, value, and user satisfaction? We ran an exploratory analysis to describe the current state of the research on empathy training in service and to answer our research questions [S1 Checklist].

### 2.2. Search strategy

To maximize the subject covering, we searched the significant databases in health, business, education, and psychology: CINAHL Plus with Full Text (EBSCOhost), Embase, Medline (Ovid), ABI/Inform Global (ProQuest), Business Source Premier (EBSCOhost), PsycINFO (Ovid) and ERIC (EBSCOhost), to which we added the multidisciplinary Web of Science Core Collection database. An information specialist developed the search strategy and adapted it for each database [S1 Appendix].

The search terms were first identified through the research team's prior knowledge of the topic, readings, and discussions in collaboration with the librarian. We applied the generic terms "training," "service employees," and "empathy" to the eight databases. We added specific terms to cover different training methods, diverse kinds of service employees, and various empathic behaviors. To do so, we applied controlled vocabulary by using a thesaurus. We completed with free text terms like training, teaching, courses, education methods, educational methodologies, experiential learning, scenario techniques, simulation, role play, role-playing, virtual reality, storytelling, service employees, service personnel, service staff, service workers, service organizations, educational services, business services, customer services, health services, caregivers, carers, therapists, counselors, doctors, healthcare providers, nurses, physicians, psychologists, empathy, altruism, compassion, sympathy, emotional intelligence, emotional connection, emotional contagion, helping behavior, helping attitude, active listening, prosocial behavior, prosocial attitude, prosocial behavior, prosocial attitude, theory of mind, understanding of others (see Table 1).

### 2.3. Inclusion/exclusion criteria and selection process of relevant articles

We included articles identified from these research databases in the final sample only if they adhered to the following search criteria: 1) the article is a qualitative, quantitative, or mixed-methods empirical study (i.e., includes qualitative and quantitative methods and data); 2) at least one training in empathy is identifiable; 3) the described training(s) in empathy is/are developed for or tested with SEs dealing with SUs. We included only papers whose research objectives explicitly stated the aim of testing a method (e.g., narrative training, role-play, mindfulness, communication training) to promote, improve, or develop SEs' empathic skills during service delivery. Articles were restricted to non-students, adult people (18+) in charge of service delivery (nursing, medicine, business, education, health).

We excluded editorials, comments, letters to the editor, and technical notes. We also excluded articles reviewing training in empathy, but we checked them for additional references. We limited our search to peer-reviewed journal articles published in English, French,

**Table 1. Inclusion and exclusion criteria for literature search.**

| CATEGORIES | INCLUSION CRITERIA | EXCLUSION CRITERIA |
|---|---|---|
| Years | 2009–2022 | |
| Keywords | Empathy and training and related terms | |
| Research settings | All settings/domains in which frontline employees deliver services to customers and/or users (nursing, medicine, business, education, health) | |
| Sample characteristics | Non-student<br>Adults (18+)<br>Frontline employees | |
| Research designs | Qualitative design<br>Quantitative design<br>Mixed methods design | Editorials, comments, letters to the editor, and technical notes<br>Reviews of empathy training<br>Reviews of the effectiveness of empathy training |
| Training | Focus: empathy training. The training is developed for or tested with frontline employees dealing with customers and/or users | |
| Outcome variables | Any empathy-related outcome | |
| Others | Published in English, French, Spanish, or Italian | |

Italian, and Spanish to narrow the literature search scope. We initially set up our systematic literature search to cover articles published within ten years (from January 1, 2009, to August 6, 2019). However, imponderable events forced us to delay the submission of our results (e.g., the Covid-19 pandemic), and we had to update our search to cover the period from August 6, 2019, to April 1, 2022. The following numbers consider this update (numbers under brackets represent the score for the first and the second batch of searches).

When applied in title, abstract, and keyword fields, the search produced 20,300 articles (15,043 + 5,257 for the update), to which we removed 6,416 duplicates (4,707 + 1,709). By screening the 13,884 references remaining (10,336 + 3548), we excluded 13,603 of them (10,075 + 3528). We assessed 281 articles for eligibility (261 + 20), eliminating those that failed to meet our inclusion criteria. The final sample was 44 relevant articles (38 + 6) (see the systematic review process flow chart in Fig 1).

A team of three reviewers screened the candidate articles and selected qualified studies independently using a multi-level title-first method [13]. After screening the title, a second (abstract) and third level (full text), screening was conducted if necessary to obtain an agreement regarding the inclusion or exclusion of the article. In case of disagreement for final inclusion, a fourth researcher assessed the inclusion eligibility of the paper after attempting to resolve it through discussion. The screening was conducted blindly using online reference management and screening tool (Covidence). Inter-rater agreement was determined by calculating Cohen's kappa coefficient.

Articles meeting the inclusion criteria were subjected to quality assessment using the Mixed-Method Assessment Tool (MMAT) [S1 Table]. The MMAT (available in S1 Table) provides a unique tool to assess the methodological quality of quantitative (experimental, quasi-experimental, and descriptive), qualitative, and mixed-methods studies based on five criteria [14]. Each criterion is rated as 'yes,' 'no,' or 'can't tell.' Two reviewers independently applied the MMAT and provided a final score based on consensus. Each 'yes' response was scored "1" while 'no' and 'can't tell' were scored "0".

## 2.4 Data synthesis approach

We used a data-based convergent synthesis design to analyze the quantitative and qualitative data [15]. We analyzed all included studies employing the same synthesis method. We

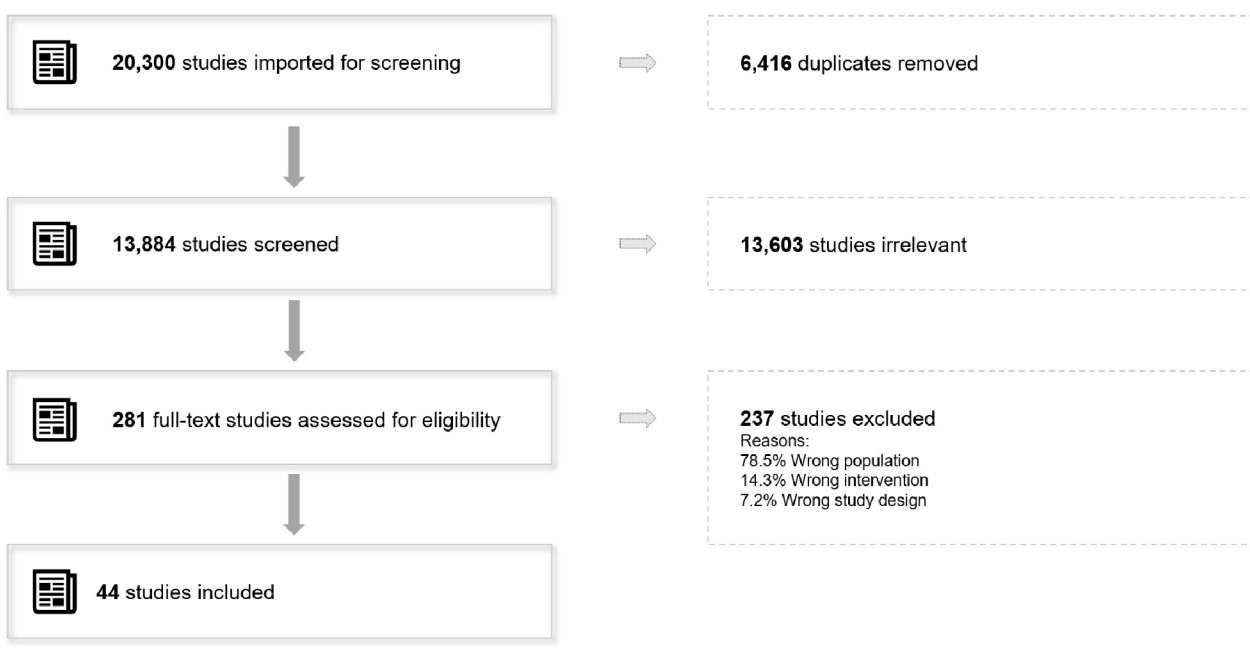

**Fig 1. Flow chart of the systematic review process.**

presented quantitative and qualitative data in the results section under a narrative presentation. We transformed the quantitative results into qualitative ones by reporting, for example, the description of the effectiveness of the empathy training programs instead of the presentation of regrouped statistics. We relied on the TIDieR checklist [11] to report the description of interventions through 6 questions: Why? Who? What? How? Where? When and how much?

## 3. Results section

### 3.1. Description of the included studies and methodological quality assessment

Results of the 44 included studies on empathy training rely on at least 6855 career professionals practicing within their field: medicine (n = 4901), nursing (n = 1145), social work (n = 322), psychological counseling (n = 82), therapy services (n = 55), other (n = 350). Twenty-six selected studies were published between 2009 and 2016, and 18 were published within the last five years (see Table 2). The selected studies have been carried out on five continents: Africa (South Africa; n = 1), Oceania (Australia; n = 3), North America (Canada, USA; n = 14), Europe (Spain, Sweden, UK, Switzerland, Germany, Italy, France; n = 16), and Asia (Taiwan, South Korea, Turkey, Japan, Iran, China; n = 10). Empathy training implementation and evaluation are somewhat biased toward Western countries (n = 30). However, there is still a significant diversity to account for cultural differences. The study design distribution is as follows: 61.4% Quantitative Non-Randomized (n = 27), 18.2% Mixed Methods (n = 8), 15.9% Quantitative Randomized (n = 7), and 4.5% Qualitative (n = 2). The selected studies are published in leading healthcare (n = 33) or psychology journals (n = 11). The leading healthcare journals include *Journal of Pediatric Nursing*, *Journal of the American Medical Association*, *Journal of Advanced Nursing*, *International Journal of Medical Sciences*, *Journal of Medical Imaging and Radiation Sciences*, and *BMC Medical Education*. The leading psychology journals include

**Table 2. Training service employees to empathic capacity: Characteristics and main findings of included studies.**

| CITATION | OBJECTIVES | DESIGN | SAMPLE | COUNTRY | METHOD | DURATION | FINDINGS |
|---|---|---|---|---|---|---|---|
| [16] | Empathy is essential and beneficial for nurses. How can it be increased? | Qualitative | N = 8 pediatric rehabilitation nurses | Canada | Arts-based narrative training intervention | Six 90-minute weekly group narrative training sessions and two in-depth interviews pre-and post-intervention. | Positive effect on empathy for patients, empathy between nurses on nursing teams, and the ability for nurses to grow increasingly more self-aware of their work's emotional and social impacts. Storytelling through narrative training may be a promising intervention tool that humanizes the clinical environment and permits nurses to share, legitimize, and make meaning of complex care experiences. |
| [17] | Empathy is essential for therapists as it facilitates their function. How do therapists respond to empathy training? | Qualitative | N = 13; 4 psychologists, 4 counselors, 1 psychotherapist, 1 social worker, 1 mental health nurse, 1 art therapist, and 1 psychiatrist. | Australia | Mindfulness practices | Six once-weekly 90-mins sessions | Enhanced empathy with customers' emotional experience, enhanced awareness of functioning as a therapist, and thoughts about how to proceed in therapy. |
| [18] | Long-term effects of a mindfulness program on burnout, mood states, empathy, and mindfulness in primary care professionals. | Quantitative non-randomized | N = 87 primary care professionals (physicians, nurses, social workers, psychologists) | Spain | Mindfulness training | 28-hrs (N = 8, 2.5-hrs weekly sessions and N = 1 8-hrs session) and N = 1 maintenance phase (N = 1 monthly session of 2.5-hrs during N = 10 months). | The scores of burnout, mood, and empathy improved significantly |
| [19] | Can empathy training reduce burnout and mood disturbance and increase empathy in healthcare professionals? | Quantitative RCT. A pragmatic randomized controlled trial with pre-and postintervention measurements | N = 68 primary health care professionals (43 in the intervention and 25 in the control group). 33.3% nurses, 60% physicians, and 6.7% social workers | Spain | (1) Educational presentation; (2) Formal mindfulness meditation; (3) Narrative and appreciative inquiry exercises; (4) Discussion | Eight sessions of 2.5 hrs per week plus a 1-day session of 8 hrs | The magnitude of the change was significant in total mood disturbance and mindfulness and moderate in burnout and empathy scales. No significant differences were found in the control group. |
| [20] | Can communications training featuring empathy improve nurse-to-patient communication? | Quantitative, non-randomized. | N = 342 inpatient nurses working in oncology | USA | Comskil training program | 1-day nursing program followed by 2-hrs modules from Jan. 2012 to Nov. 2014. | Nurse participants' self-efficacy in responding empathically to patients significantly increased. |

(*Continued*)

**Table 2.** (Continued)

| CITATION | OBJECTIVES | DESIGN | SAMPLE | COUNTRY | METHOD | DURATION | FINDINGS |
|---|---|---|---|---|---|---|---|
| [21] | What is the impact of empathic communication skills training on reducing lung cancer patients' experience of stigma? | Quantitative non-randomized | N = 30 HCPs working in oncology | USA | Communication Skills Training and Research Lab (Comskil) | 2.25 hours session conducted from December 2017 to April 2019 | The HCPs favorably received the training and have the potential to improve the patient experience. |
| [22] | How do psychologists and other staff respond to a proposed emotional literacy program? | Mixed-Methods, a pre-experimental research design, one-group pre-test, and post-test approach | N = 14; 2 psychologists, 1 nursery school administrator, 3 teachers, 8 auxiliary social workers | South-Africa | Emotional Literacy and Persona Doll program. | 8-weeks training program | No significant improvement in empathy. However, the qualitative study showed that participants were more able to connect with their own emotions and the emotions of others. |
| [23] | Effect of experiential relationship-centered physician communication skills training on patient satisfaction and physician experience. | Quantitative non-randomized, observational study | N = 1537 physicians who participated in, and N = 1951 physicians who did not join in, communication skills training. | USA | Communication skills training | One 8-hrs block of interactive didactics live or video skill demonstrations, and small group and large group skills practice sessions using a relationship-centered model between Aug. 2013 and Apr. 2014. | Overall, Clinician and Group Consumer Assessment of Healthcare Providers and Systems scores for physician communication were higher for intervention physicians than for controls. Significant improvement in the post-course Hospital Consumer Assessment of Healthcare Providers and Systems. Physicians showed significant improvement in empathy and burnout, including all measures of emotional exhaustion, depersonalization, and personal accomplishment. Less depersonalization and greater personal accomplishment were sustained for at least three months. |

*(Continued)*

**Table 2.** (Continued)

| CITATION | OBJECTIVES | DESIGN | SAMPLE | COUNTRY | METHOD | DURATION | FINDINGS |
|---|---|---|---|---|---|---|---|
| [24] | Effect of communication training on physician-expressed empathy using two measures (global and hierarchical) of physician empathic behavior. | Quantitative RCT | N = 232 audiotaped physician-patient interactions. | USA | Communication training on physician-expressed empathy | Three 6-hrs communication skills-based sessions. Physicians also received a 30–45 min individual coaching session after each workshop that included a review of an audiotape of a recent patient visit in 11 months. | The differences in global empathy scores in the physician training group from baseline to follow-up improved by 37%, and hierarchical scores of physician empathic expression improved by up to 51% from baseline scores for the same group. |
| [25] | Effect of a course in communication on the content of nurse–parent encounters and the ability of nurses to respond to the empathic needs of parents in a level III neonatal intensive care unit. | Quantitative non-randomized | N = 13 nurses | Sweden | Communication training | 2 hrs interactive lecture and one-day practical workshop. | The use of empathic or exploring responses to empathic opportunities increased, whereas ignoring the parents' feelings or giving inadequate advice decreased after the course.<br><br>The use of statements expressing caring for the parents and encouraging parents to participate in the care of their infant increased after the course. |
| [26] | Effect of radiation therapists' techniques for assisting patients experiencing treatment-related anxiety. | Mixed-Methods | N = 12 radiation therapists | Canada | Radiation therapists (RTs) techniques | 8 weeks on-the-job training program | No change in the perceived stress before and after training. Qualitative findings showed significant benefits to RTs, including (1) the ability to empathically attune more effectively and earlier to signs of anxiety in patients; (2) improved confidence and self-efficacy for effectively intervening in difficult treatment situations; and (3) enhanced creative problem-solving in partnership with patients to assist the acutely anxious patient. |

(*Continued*)

**Table 2.** (Continued)

| CITATION | OBJECTIVES | DESIGN | SAMPLE | COUNTRY | METHOD | DURATION | FINDINGS |
|---|---|---|---|---|---|---|---|
| [27] | How does the NM (Narrative Medicine) program impact the empathy scores of healthcare providers? | Quantitative non-randomized | N = 142 multi-professional healthcare providers (n = 122 females) were divided into single (n = 58) and team groups (n = 84) on the basis of inter-professional education | Taiwan | Narrative Medicine (NM) program | Two months | Empathy scores increased after the NM program and were sustainable for 1.5 years for all participants.<br><br>No significant effect of gender over time was found, but a trend showed females increasing empathy scores at T2, sustaining at T3, but males demonstrating a slow rise in empathy scores over time. |
| [28] | Effect of an educational program for patient and family advisors' collaboration on empathy levels of intensive care unit nurses. | Quantitative non-randomized | N = 30 nurses | USA | Patient-Family Advisors designed educational interventions using simulation-based role-playing. | 4-hrs class; once | Empathy scores significantly increased after nurses completed the PFA-designed educational program.<br><br>For the changes in TEQ scores from preintervention to postintervention, age was significantly associated with improvements in TEQ scores. |
| [29] | Effect of real-world trauma center providers training on providing higher quality counseling using Motivational Interviewing as part of brief interventions for alcohol and whether Motivational Interviewing skills can be maintained over time. | Quantitative RCT | N = 40; 19 nurses, 15 social workers, 4 physicians assistants, 1 chemical dependency professional, 1 respiratory therapist | USA | Motivational interviewing | 1-day on-site workshop, 27 months | Improved Motivational Interviewing skills scores throughout the 27 months (spirit and empathy); however, despite the overall improvement. |
| [30] | Effect of Self-practice/self-reflection on technical and interpersonal skills. | Quantitative non-randomized, in the context of a quasi-experimental design including multiple baselines within a single-case methodology | N = 14 Experienced cognitive-behavioral therapists | United-Kingdom | Self-practice/self-reflection (SP/SR) in an experiential cognitive-behavioral therapy (CBT) training program | Ten weeks | SP/SR enhances both technical and interpersonal therapeutic skills.<br><br>Self-perceived empathy skills were significantly higher post-SP/SR than pre-SP/SR for those participants who completed the program. |

(*Continued*)

**Table 2.** (Continued)

| CITATION | OBJECTIVES | DESIGN | SAMPLE | COUNTRY | METHOD | DURATION | FINDINGS |
|---|---|---|---|---|---|---|---|
| [31] | Effect of an educational intervention on nurses' attitudes towards and confidence in providing family care and families' perceptions of support from nurses in an adult critical care setting. | Mixed-methods, pilot study | N = 30 nurses | USA | Educational protocol for developing therapeutic conversations with families. | A 4-hrs workshop | Increased confidence, knowledge, and skill following the educational intervention. |
| [32] | Effect of Motivational Interviewing on inspectors' skills to promote environmentally sustainable behavior in inspectees. | Quantitative non-randomized | N = 40 inspectors | Sweden | Motivational interviewing | 6-days | Increased competence in empathy |
| [33] | Assess the feasibility and usefulness of adherence counseling training, skills, and support tools for GPs. | Quantitative non-randomized | N = 25 general practitioners (GPs) | Australia | Counseling training | 2-hrs training workshop, 6-months | GPs' confidence in using counseling skills increased, as did the frequency they applied the skills and their satisfaction with consultations.<br><br>Patients reported good GP empathy and no significant change in adherence barriers |
| [34] | Effect of physician training in empathic skills on patients' satisfaction after their first consultation in a private fertility clinic setting. | Quantitative non-randomized | N = 30 physicians | Spain | Empathic training | 2-days | Increased empathy. |
| [35] | Effect of a course on communication skills training based on counseling strategies for a group of seven fellows of nephrology. | Quantitative non-randomized | N = 8 fellows in nephrology | Spain | Communication skills training | 2-hrs training, 8-monthly sessions | The frequency of spontaneous empathic responses rose.<br><br>The level of perceived competence to face difficult situations increased.<br><br>Subjective perception of patient involvement also increased |

(*Continued*)

**Table 2.** (*Continued*)

| CITATION | OBJECTIVES | DESIGN | SAMPLE | COUNTRY | METHOD | DURATION | FINDINGS |
|---|---|---|---|---|---|---|---|
| [36] | Effect of a mindfulness training program on the levels of burnout, mindfulness, empathy, and Self-compassion among Healthcare professionals in an Intensive Care Unit of a tertiary hospital | Quantitative non-randomized | N = 22; 3 nurses, 11 nursing assistants, 8 physicians | Spain | Mindfulness training | 8-weeks individual training, N = 1 plenary session | Emotional exhaustion decreased, and self-compassion levels increased. Empathy and Mindfulness levels were not globally modified |
| [37] | The effect of two empathy enhancement programs on direct care workers of older adults living alone. | Quantitative non-randomized | N = 104 direct care workers, 52 in each group | South Korea | Empathy training | 50 min for simulation, 30 min for lecture | Only the lecture group reported significantly higher empathy levels. |
| [38] | How does a simulation-based empathy program impact the empathy skill of care providers for older adults? | Mixed-methods | N = 104 care providers | South Korea | Empathy training simulation | 50 minutes session | The post-test revealed a significant increase in empathy and compassion fatigue resistance following the simulation training |
| [39] | Effect a digital training intervention, 'In Their Shoes,' which immerses participants in the experience of living with inflammatory bowel disease (IBD), highlighting the biopsychosocial impact. | Mixed-methods | N = 104 employees | United Kingdom | Empathic training "In Their Shoes." | 36-hrs training | Increases in IBD understanding and connection to patients, evaluation of organizational innovation, empathy, and prosocial job perceptions. |
| [40] | Can the attitude of health staff towards those with intellectual disabilities be improved through empathy training? | Quantitative non-randomized | N = 76 health service staff | United Kingdom | Training package about the experiences of individuals whose behavior challenges–Who's Challenging Who (WCW). | 3-hrs and 20 mins training, N = 10 WCW training sessions | Significant positive change in attitudes, empathy, and self-efficacy. Larger changes in empathy were found for older staff and staff who had been working in health and social care for longer |
| [41] | Effect of empathy training on the empathic skills of nurses | Quantitative non-randomized | N = 48 nurses | Turkey | Empathy training | 20-hrs, N = 5 training sessions | Increase in empathic skills for the experimental group; no significant increase for the control group. |
| [42] | Develop empathic approaches in the nurses who care for adolescents with type 1 diabetes mellitus. | Quantitative non-randomized | N = 8 nurses;<br><br>N = 46 adolescents with diabetes mellitus | Turkey | Empathy training | 16-hrs, N = 2 days | Nurses use more empathic behaviors after than before the training. |

(*Continued*)

**Table 2.** (Continued)

| CITATION | OBJECTIVES | DESIGN | SAMPLE | COUNTRY | METHOD | DURATION | FINDINGS |
|---|---|---|---|---|---|---|---|
| [43] | Effect of an intensive educational program in mindfulness, communication, and self-awareness on improvement in primary care physicians' well-being, psychological distress, burnout, and capacity for relating to patients. | Quantitative non-randomized | N = 70 primary care physicians | USA | Mindfulness training | 8-weeks intensive phase (2.5 hrs/wk., 7-hrs retreat) followed by a 10-months maintenance phase (2.5 hrs/mo.) | Improvements in mindfulness, burnout (emotional exhaustion), personal accomplishment, empathy, physician belief, mood disturbance, and personality (conscientiousness and emotional stability). |
| [44] | What is the effect of the multimodal comprehensive care methodology program, "Humanitude," on empathy towards people with dementia among oral care professionals? | Quantitative non-randomized | N = 45 dentists and oral hygienists | Japan | comprehensive care methodology training program | 7 hours session | The empathy scores toward patients with dementia increased significantly following the program. Patients' dental health also saw improvement based on the oral health assessment tool. |
| [45] | Effect of a communication skills training program for oncology nurses. | Quantitative non-randomized | N = 70 nurses | Switzerland | Communication skills training program | 2,5 days and 1 booster session (6 months after the initial session) that lasted 1,5 days | Increase in appropriate empathic, reassuring statements and questions concerning psychosocial information; utterances containing medical information decreased on the part of nurses and patients; and patients provided more psychosocial information.<br><br>The level of congruence and empathic responses to patients' emotional cues increased, as did the length of uninterrupted speech |
| [46] | Describe burnout, empathy, and satisfaction at work and explore whether a tailored program based on motivational interviewing techniques modifies and improves such features. | Quantitative non-randomized | N = 45 professionals working in a spinal cord injury unit | Spain | Motivational interviewing (MI) | 12-hrs, 2-days training | Professionals are performing quite well, and they refer to satisfactory empathy, satisfaction at work, and no signs of burnout or significant stress both before and after the training |

(*Continued*)

**Table 2.** (Continued)

| CITATION | OBJECTIVES | DESIGN | SAMPLE | COUNTRY | METHOD | DURATION | FINDINGS |
|---|---|---|---|---|---|---|---|
| [47] | Effect of empathy in child-care professionals (i.e., teachers, psychologists, social workers) in preventing sexual abuse against children and youngsters. | Quantitative non-randomized | N = 42/94 experienced professionals working with children | Germany | E-learning Empathy training | 6-months | Significant progress was found in Situational Empathy scales and some Coping subscales. The outcomes indicate that the prevention program elicits important changes in the cognitive sphere and that these changes are more intense when the implication level for the situation is greater. This research shows that empathy can be improved through professional experience and careful situational involvement. |
| [48] | Effect of a three-day residential course concerning empathy and counseling abilities on patients' ratings of the level of empathy of physicians and nurses working in vaccination centers. | Quantitative non-randomized | N = 19; 11 nurses, 8 doctors | Italy | Empathy training | 18 hrs on three days; 4 sessions during 4 or 5 hours each | Increase empathy and counseling skills |
| [49] | Effect of a communication skills training program on emergency nurses and patient satisfaction. | Quantitative non-randomized | N = 16 nurses, | Turkey | Communication skills training program | Six weeks, 90 mins per week | Empathy and communication skill scores increased after training.<br><br>The patient satisfaction survey of 429 patients showed increased scores on confidence in the nurses; the nurse's respect, kindness, and thoughtfulness; individualized attention; devotion of adequate time to listening; and counseling and information delivery. The number of undesirable events and complaints during nurse-patient interactions decreased significantly. |
| [50] | Determine the effectiveness of empathy training on the empathy skills of nurses working in intensive care units. | Quantitative randomized | N = 80 Nurses | Iran | Empathy training | Eight sessions in 90 minutes | The results indicate a significant increase in empathy scores. |

(*Continued*)

**Table 2.** (Continued)

| CITATION | OBJECTIVES | DESIGN | SAMPLE | COUNTRY | METHOD | DURATION | FINDINGS |
|---|---|---|---|---|---|---|---|
| [51] | Explore how guided reflective writing could evoke empathy and reflection in a group of practicing physicians. | Mixed-methods | N = 40 physicians | USA | Reflective writing training | N = 6 sessions (Sessions 1 and 6 were 4-hrs in duration, and the remaining sessions were 2-hrs each) | Qualitative analysis of physicians' writings showed compassionate solidarity and detached concern themes. Exploration of negative emotions occurred more frequently than positive ones. The most common writing style was case presentation.<br><br>Results of statistical analysis suggested an improvement in empathy in the intervention group at the end of the course |
| [52] | Selection of a multidimensional approach towards empathy to determine whether cognitive and emotional aspects might differently contribute to burnout (first study). A second study investigated the effect of an empathy-based training program on empathic skills and burnout. | Quantitative non-randomized | N = 124 nurses | France | Empathy training | 3 days | Results showed that higher personal distress predicted higher burnout scores, while higher compassionate care predicted lower emotional exhaustion and higher perspective-taking predicted lower depersonalization and higher accomplishment.<br><br>Results from the second study showed that personal distress decreased after the validation training, and nurses reported lower depersonalization and higher accomplishment |
| [53] | Determine whether the SymPulseTM device could enhance feelings of empathy in test participants (wearing the device) versus control participants (not wearing the device). | Quantitative RCT | N = 45 participants | Canada and USA | Digital tele-empathy device for use toward patients with Parkinson's disease | Once | Scores for trait empathy revealed no significant difference between test and control participants. By contrast, scores for state empathy revealed significantly higher test scores than control participants. |

(*Continued*)

**Table 2.** (Continued)

| CITATION | OBJECTIVES | DESIGN | SAMPLE | COUNTRY | METHOD | DURATION | FINDINGS |
|---|---|---|---|---|---|---|---|
| [54] | Report on developing, implementing, and evaluating a Communication Skills Training module for inpatient oncology nurses on empathizing with patients. | Quantitative non-randomized | N = 248 nurses | USA | Communication skills training program | 1-day | Nurses' self-efficacy in responding empathically significantly increased. Nurses showed improvement in empathy skills and reported feeling confident in using the skills they learned post-training and reported an increase of 42–63% in using specific empathic skills. |
| [55] | Report the development and implementation of simulation-based empathic communication training and the evaluation of training efficacy | Quantitative non-randomized | N = 32 nurses | China | Simulation-based empathic communication training | Four weeks, once a week | Nurses' self-reported attitude and confidence concerning their empathy skills and understanding of empathic communication reflected significant improvement. The behaviors of nurses toward communicating empathetically improved significantly after undergoing the simulation training |
| [56] | This study compares the learning outcome of virtual reality empathy workshops versus non-virtual reality empathy workshops using dementia care workers. | Quantitative randomized | N = 114 care workers | Australia | Virtual reality empathy workshop | 3 hours | Empathy and understanding of dementia patients increased significantly in both conditions. Virtual reality training had a stronger effect on older and younger participants. |
| [57] | Effect a brief, computerized intervention on oncologist responses to patient expressions of negative emotion. | Quantitative RCT | N = 48 oncologists | USA | Empathy training | N/A | Oncologists in the intervention group used more empathic statements and were more likely to respond to negative emotions empathically than control oncologists. Patients of intervention oncologists reported greater trust in their oncologists than those of control oncologists. There was no significant difference in perceptions of communication skills. |

(*Continued*)

**Table 2.** (Continued)

| CITATION | OBJECTIVES | DESIGN | SAMPLE | COUNTRY | METHOD | DURATION | FINDINGS |
|---|---|---|---|---|---|---|---|
| [58] | Effect of a psychosocial training program for speech therapists on their performance skills in patient-therapist communication in general and empathy in particular. | Mixed-Methods | N = 32 speech therapists in oncology | Germany | Psychosocial training programme | Four 2-days training units | Communication skills improved considerably regarding the frequency of conducive communication (especially empathy) and the width of conducive communicative repertoire. Negative communication preferences were reduced. |
| [59] | Investigate whether physicians' intrapersonal empathy increased after a communication skills training workshop. | Quantitative non-randomized | N = 507 oncologists | Japan | Communication skills training workshop | One hour for a total of 2 days | Empathy scores increased significantly. Perspective-taking and empathic concern scores increased significantly, whereas fantasy and personal distress scores showed no significant changes. The scores of palliative care physician participants increased significantly. |

*Frontiers in Psychology*, *Journal of Clinical Psychology and Psychotherapy*, and *Journal of Counselling and Psychotherapy Research*.

There were 7 (15.9%) experimental and 28 (63.6%) non-experimental quantitative studies. Qualitative and mixed-methods designs represented 4.5% (n = 2) and 15.9% (n = 7) of the included studies. Most experimental studies had a good to excellent MMAT score (mean of 3.9 out of 5), whereas quasi-experimental studies had a lower MMAT score (mean of 3/5). The two qualitative studies had an excellent MMAT score (5/5), and the mixed-methods studies had more variable MMAT scores, with means of 3.1/5 for the quantitative part, 4.7/5 for the qualitative part, and 2.3/5 for the mixed-methods component. The details of the MMAT score for each study are presented in the S1 Table. The most poorly reported criteria are given here per each design. Regarding the experimental studies, the criterion was: 2.1. Is randomization appropriately performed? For the quasi-experimental studies, those two criteria were: *3.1 Are the participants representative of the target population*? *3.4 Are the confounders accounted for in the design and analysis*? Regarding the mixed-methods studies, the two criteria that were not consistently reported were: *5.2. Are the different components of the study effectively integrated to answer the research question*? *5.4. Are divergences and inconsistencies between quantitative and qualitative results adequately addressed*?

## 3.2. Why?

The main goal of empathy training is to increase SEs' empathy toward SUs. But how researchers define empathy leads to specific behaviors targeted and facilitated through empathy training. Only half of the selected studies (n = 21) define empathy (see Table 3), which is mainly viewed as a cognitive rather than an affective response [i.e., SEs' ability to *understand* SUs' emotions; 38]. This cognitive framing denotes a practical approach to empathy training aimed

**Table 3. Definitions of empathy in the selected studies.** 21 papers out of the 44 selected papers did not define empathy.

| CITATIONS | DEFINITIONS |
|---|---|
| [16] | The capacity to imagine the situation of each patient and their family—understanding their feelings and perspective, and responding in ways that make patients feel heard and cared for. |
| [22] | Empathy concerns the ability to take another's perspective or point of view and the ability to transpose one's self into the situation of others. Empathy also incorporates the ability to have "other-oriented feelings" and concern, as well as anxiety or distress in interpersonal situations. |
| [27] | Physician empathy is a multidimensional concept involving cognitive and affective domains. The former involves the ability to understand another person's inner experiences and feelings alongside a capability to view the outside world from the other person's perspective. The latter involves the capacity to enter into or join the experiences and feelings of another person. |
| [32] | Accurate listening to inspectees. |
| [34] | Empathy, in the case of a medical relationship, should be the feeling, by the patient, of being understood and accepted by the physician. Empathy encompasses a cognitive aspect and an affective one; cognitive empathy allows the physician to apprehend the point of view of the patients and establish effective communication, and affective empathy lets him or her respond to their emotional state, creating a partnership within which interpersonal trust develop. |
| [39] | Empathy can be described as having two inter-related dimensions: cognitive and affective. Cognitive empathy measures the skills-based aspect of learning, where a person is able to recognize and understand another's experience. Then affective empathy links to the transformative aspect of the learning cycle, where the understanding resonates emotionally with the individual and they start to be able to interpret their knowledge, exploring concepts beyond the facts they are presented with. |
| [37, 38] | Empathy refers to an understanding of experiences, concerns, and perspectives of clients, the ability to communicate this understanding, and the intention to help. |
| [41] | Empathy is defined as the ability to understand how others feel and what they mean, and to convey these emotions to others. It is currently believed that empathy is multi-dimensional and involves emotional, cognitive, communicative, behavioral, moral, and relational dimensions. |
| [42] | Empathy represents the similarities in experiences and the wish to understand others. |
| [44] | Empathy is defined as the ability to understand a patient's experiences and feelings, as well as the ability to communicate this understanding. |
| [46] | Empathy is the ability of understanding patients' feelings and concerns and it has been related to an increased likelihood of patients' adherence to treatment. |
| [47] | An integrative approach to empathy encompasses multiple dimensions, such as affective sharing, awareness of self and others, emotional regulation, perspective-taking, and empathy-related responding. |
| [48] | Being interested in parents and children as whole people, listening to them, and understanding their expectations and concerns about vaccination. |
| [50] | Empathy is the skill of perceiving the feelings and views of others and the means to effectively communicate with the patient. |
| [51] | Empathic communication is defined as the skill of understanding the patient's perspective. |
| [52] | Through the relation to self, empathy is defined as the capacity to understand patients' emotions and to share that understanding with the patient, which promotes helping behaviors. |
| [53] | Clinical empathy involves both cognitive and affective components, which include (1) understanding the patient's situation, thoughts, and feelings, (2) verifying its precision with the patient, and (3) responding to the patient in a helpful manner. |
| [54] | Empathy is a two-stage process: (1) the understanding and sensitive appreciation of another person's predicament or feelings and (2) the communication of that understanding back to the patient in a supportive way. |
| [55] | Empathy is defined as a two-phase process: (a) understanding and appreciating another person's feelings and emotions and (b) communicating understanding back to the patient in a supportive way. |
| [58] | Empathy reflects needs and emotions and uses the patient's language and metaphors. |
| [59] | Empathy refers to at least three qualities: (1) the emotional dimension (i.e., feeling what another person is feeling), (2) the cognitive dimension (i.e., understanding what another person is feeling), and (3) the behavioral dimension (i.e., responding compassionately to the distress of another person). |

at detecting and using SU's emotions as social information to adapt the service delivery. For instance, empathy in rehabilitation nursing refers to *understanding* each patient's situation while responding caringly [16]. Therefore, empathy training focuses on storytelling exercises that challenge SEs to identify, describe, and create emotional narratives. In medicine, empathy is a *feeling of understanding* between the physician and the patient to develop a trusting partnership [34]. Therefore, empathy training relies on four methods—case study, role-playing, active listening, and social style identification—to develop physicians' cognitive (i.e., perspective-taking) and affective (i.e., emotion sharing) empathy in interpersonal physician-patient relationships [34].

Independently on specific service settings, empathy training goals can be further divided into four categories: (1) communication skills (n = 21; 47.7%); (2) relationship building (n = 15; 34.1%); (3) emotional resilience (n = 6; 13.6%); and (4) counseling skills (n = 2; 4.5%). Communication skills allow the effective sharing of information between parties [e.g. 20, 35] and apply to service interactions that rely on frequent and sensitive interactions [23, 57]. Relationship building aims at helping SEs to form effective bonds with SUs through empathy [e.g., 28, 41] and apply to service settings that require prolonged interactions with vulnerable SUs [16, 38]. Emotional resilience aims at providing SEs with the necessary self-regulation skills to reduce professional and emotional burnout [e.g., 19, 23, 36] and apply to demanding and emotionally exhaustive service settings [e.g., 38, 43]. Finally, counseling skills empower SEs to support SUs through strategies such as motivational interviewing [e.g., exploring SU's perspective; 29, 30].

The purpose of empathy training could also be classified according to the beneficiary—the SEs (e.g., reducing burnout) or the SUs (e.g., increasing service satisfaction). Empathy training to improve SUs' service satisfaction is the most prevalent among the selected studies (n = 28; 63.6%) [e.g., 20, 25]. For instance, training can help SEs with frequent and sensitive interactions with SUs (e.g., therapy) improve service quality through empathic communication [e.g., 35, 44]. Then, selected studies that target SEs' benefits through empathy training (n = 16; 36.4%) aim to reduce SEs' burnout and emotional exhaustion, especially among those working in intensive positions, such as Intensive Care Unit nurses or primary care physicians [e.g., 36, 43]. Other desired employee-centric benefits include increased self-compassion and confidence, which aid SEs in coping with the demands of their job [e.g., 31], and changed job perception and increased prosocial attitudes toward their role [e.g., 16, 39].

## 3.3. Who?

The 44 selected studies focus mainly on adult professionals working in health-adjacent fields: Nurses (n = 13; 29.5%), physicians (n = 13; 29.5%), mental health professionals (n = 6; 13.6%), therapists (n = 4; 9.1%), care workers (n = 3; 6.8%), and others (n = 5; 11.4%). The combined number of participants in empathy training totalized 6855, with the following occupational breakdown: Physicians (71.5%), nurses (16.7%), care workers (4.7%), mental health professionals (1.2%), therapists (0.8%), and others (3.4%). Physicians account for a disproportionate number of respondents due to a few studies using large sample sizes [e.g., 23, 59]. Also, the overrepresentation of professional health workers in empathy training programs is coherent with professional requirements in the health sector. To communicate with patients about sensitive topics such as diagnosing lung cancer, physicians' ability to empathize is essential [23]. Empathy training can help improve the outcome of these stigmatized communications [21]. The role of nurses, especially in sensitive sectors such as neonatal care [25], is also highly dependent on effective and compassionate communication during distressing situations [20]. Finally, mental health workers, caregivers, and therapists benefit greatly from emotional

literacy and empathy training programs to build strong relationships with their clients through effective and empathic communication [e.g., 22], such as understanding the difficulties experienced by those who have dementia [56].

### 3.4. What?

The content brought forward through empathy training varies greatly depending on the context of the profession towards which the training is applied. For instance, care workers working with dementia patients are introduced to an empathy training session simulating the debilitating effects of dementia [37]. Empathy training for dental care professionals working with elders relies on the "*Humanitude*" method, which focuses on communication skills through gaze, talk, touch, and assistance with standing up [44]. Empathy training can also contain a role-play module (taking the role of the SU) to foster empathic understanding [17] or a narrative approach to humanize the environment for SEs through sharing and interpretation of stories [e.g., 16, 27]. Another empathy training program consists of physicians receiving scenario-based modules called "*In Their Shoes*," detailing the challenges of patients living with inflammatory bowel disease to understand their perspective better [39].

### 3.5. How?

Empathy training programs are mainly carried out face-to-face (n = 22; 50%), in group settings (n = 12; 27.3%), or distance learning (n = 3; 6.8%) (others: n = 7; 15.9%). Face-to-face and group-setting interventions are very common. They allow for interpersonal exercises, such as role-play [e.g., 28], and specific methods, such as the "Comskil" program to improve empathic skills through seven modules [20, 21]: (1) agenda setting, (2) questioning and history taking, (3) recognizing or eliciting a patient's empathic opportunity, (4) working toward a shared understanding of the patient's emotion/experience, (5) empathically respond to the emotion or experience, (6) facilitate coping and connect to social support, and (7) closing the conversation. Conversely, the more unusual approaches to empathy training are found in the "other" category. For instance, simulation-based empathy training exposes care workers to the challenges of living with dementia and uses sensory deprivation to emulate the condition [37].

The staff who administer the empathy training programs are research team members (n = 16; 36.4%), medical professionals (n = 12; 27.3%), trainers (n = 8; 18.2%), and other professionals (n = 8; 18.2%). The more straightforward and less engaging empathy training formats are often led by untrained staff or research team members. These simple-to-run training formats include educational videos and simulations that require minimal involvement from the provider [e.g., 38]. The structured empathy training sessions are also provided by existing employees hosting the training [e.g., 56]. The more complex training sessions usually involve dynamic exercises, such as expressive writing workshops and role-play, requiring professional staff to facilitate them [e.g., 16].

### 3.6. Where?

Empathy training programs are not resource-intensive and do not require much space. In the 44 selected studies, empathy training sessions took place primarily in hospitals/clinics (n = 23; 52.3%), universities/research labs (n = 7; 15.9%), and other locations (n = 14; 31.8%), such as at home using a computer program or a workbook [30, 57]. Since most research participants were employed in the health field, it is understandable that the empathy training would be conducted on-site at the hospital/clinic where they are used.

## 3.7. When and how much

The empathy training tested in the 44 selected studies followed varying timeframes: Less than a week (n = 17; 38.6%), a month or less (n = 4; 9.1%), six months or less (n = 17; 38.6%), more than six months (n = 3; 6.8%), and other (n = 3; 6.8%). The training timeframe is primarily tied to the participating workers' availability and the desired sample size [e.g., 23]. It can be administered in single or multiple sessions over time [24]. Therefore, a relevant metric to consider is the duration of the intervention itself. However, there is considerable variation among all the empathy training, demonstrating no standard duration. The breakdown is as follows: 50 minutes (n = 2), 1 hour (n = 1), 1.5 hours (n = 1), 2 hours (n = 1), 2.15 hours (n = 1), 3.3 hours (n = 1), 4 hours (n = 2), 6 hours (n = 1), 7 hours (n = 1), 8 hours (n = 1), 9 hours (n = 3), 12 hours (n = 1), 14 hours (n = 1), 16 hours (n = 2), 18 hours (n = 2), 20 hours (n = 1), 28 hours (n = 1), 36 hours (n = 2), 52 hours (n = 1), and eight sessions (n = 1). For instance, a study seeking an intensive intervention schedule spread 28 hours of course time over eight weeks [18]. Other studies include role-playing components as part of the intervention, resulting in longer sessions than the more straightforward lecture-based approach [28, 37]. The more unique training methods increase the variability of the time required. For instance, simulation-based dementia training requires only 50 minutes, including equipment setup [37]. Contrarily, an online simulation seeking to immerse the user in the daily challenges of living with inflammatory bowel disease had 36 hours of content [39].

## 3.8. Effectiveness of training

The systematic review shows that 68.2% (n = 30) of the 44 selected studies report a significant increase following the intervention in empathy scores, such as perspective-taking (i.e., cognitive empathy) or empathy-based service skills, such as accurate listening [e.g., 20, 32, 43, 54, 55]. For instance, studies focusing on improving empathic responses and communication frequency while working with SUs reported favorable results; for example, nurses working with diabetic adolescents more frequently responded empathically following a two-day empathy program [42]. Similarly, nephrology residents exposed to communication training had an increased rate of spontaneous empathic responses [35]. Other studies indicated a decrease in un-empathic communication following empathy training, further supporting the claim that empathy training effectively improves communication [e.g., 25, 58]. Evidence also confirms the direct effect of empathy training on service quality. For instance, patient satisfaction increases when interacting with nurses [34, 49, 31] or physicians [23] who have improved their empathic communication skills through empathy training programs. Empathy training also increased the service quality of Cognitive Behavior Therapy by developing therapeutic skills using self-reflection and self-practice [30].

The effect of empathy training on the well-being of SEs is also well supported by the results in the 44 selected studies. Studies show that SEs improved their ability to cope with emotional distress [e.g., 16, 18, 19, 52] and mitigate professional burnout associated with high-stress positions such as nursing following empathy training [e.g., 23, 36, 43]. This improvement in well-being extends past the individual since empathy training enhances the sense of community among the nursing staff [16].

## 3.9. Summary of findings

Consistent with our inclusion criteria, all selected studies' goal of empathy training was to increase/improve SEs' empathy toward SUs. The 21 studies that reported a definition of empathy focus mainly on the cognitive rather than the affective dimension of empathic skills, denoting a practical approach to empathy training aimed at detecting and using SU's emotions as

social information to adapt the service delivery. The results of the 44 selected studies confirm that empathy training enhances SEs' empathic skills and SUs' satisfaction. Significantly, empathy training improves SEs' empathy along four primary skills: communication skills, relationship building, emotional resilience, and counseling skills.

The results of the 44 selected studies focus mainly on adult professionals working in health-adjacent fields. However, the content brought forward through empathy training varies greatly depending on the professional context. Empathy training sessions are mainly conducted face-to-face, in group settings at the practice location (e.g., hospitals, universities), or in distance learning. Finally, the empathy training tested in the 44 selected studies followed varying time-frames in single or multiple sessions over time, demonstrating no standard duration, and seems primarily tied to the participating workers' availability and the desired sample size.

## 3.10. Limitations of the selected papers

The selected studies present some limitations (see Table 4). We already outlined the lack of definition for empathy (21 out of 44 studies) that could hinder the proper development of empathy training programs, such as module development, targeted skills, and efficiency measurements. Another primary limitation refers to sample characteristics. Most of the selected studies rely on small, almost exclusively female nurses and physicians' samples, limiting the generalizability of the results to other service domains. The participants are mainly self-selected in hospital settings, and previous experience and training in empathy are not

**Table 4. Main limitations of the 44 selected studies.**

| LIMITATIONS | EXAMPLES |
|---|---|
| Small samples produced limited analysis and results | The study sample was not large enough to conduct investigations to determine the influence of physician or patient characteristics on the effects of the intervention [24; See also: 16, 55]. |
| Participants were almost exclusively female | The participant pool had a higher proportion of females than males. Hence, the results might only be generalizable to a female population [27; See also: 19, 47]. |
| A limited population (primarily nurses and physicians) may limit generalizability | The study results were generalizable only to the nurses who provided care to adolescent diabetic patients at the adolescent service of a university hospital [42: See also 20, 46]. |
| Limited settings (mostly one hospital setting) may limit generalizability | The study was carried out at one cancer center in the northeast USA, and the results may not be generalizable to other cancer hospital settings [20; See also: 28, 51]. |
| Studies relied heavily on self-administered questionnaires | The improvement of empathic skills may be due to a Hawthorne effect, whereby the physicians may have changed their attitudes in response to being observed rather than because of the training [34; See also: 25, 40]. |
| Lack of follow-up measures | Focused only on the training program's immediate impact, so could not determine actual behavior change or longer-term effects of the intervention [39; See also: 45, 53]. |
| Some baseline measures were heterogeneous or not taken | As a pretest-post-test design, there is no way of judging whether the pretesting process influenced the results because there was no baseline measurement [34; See also: 58]. |
| Previous experience and training should be accounted for when designing the study | Participants' prior experience with communication skills training was not assessed [20; See also: 26, 29]. |
| Self-selected sampling was used often (non-randomized trials) | Participants were self-selected or designated by a nurse leader for specific reasons which were not described [20; See also: 24, 43]. |
| Measures of actual performance are lacking | Authors could not track how changes in self-report measures affected actual clinical care [43; See also: 46, 54]. |

controlled. Besides, empathy training efficiency evaluation relies on self-report measures subjected to biases such as social desirability and does not necessarily target the actual empathy performance. There is a lack of follow-up measures that could appraise the effect of empathy training over the long term. Finally, the methods, contents, and modules implemented in the empathy training programs are unclear, limiting the possibility of replicating the training in other service settings.

## 4. Discussion and research agenda

Empathy plays a crucial role in delivering a successful service experience, as supported by references to empathy in academic papers and the industry's growing emphasis on fostering empathy in every service interaction. Consequently, the significance of empathy training is acknowledged for equipping managers and service employees (SEs) with the necessary qualities of care and compassion to effectively cater to service users (SUs) during interactions [e.g., 3]. Our systematic review aimed to identify, synthesize, and discuss empirically tested empathy training in the service sector while highlighting critical areas for future research. We meticulously analyzed 44 empirical papers published between 2009 and 2022, providing a comprehensive account of how empathy training was implemented and extensively tested. However, despite repeated calls from industry and academia for empathy training, our findings revealed a lack of interest among service scholars in developing and evaluating empathy training programs for SEs. Notably, the papers we identified exclusively focused on health services and primarily involved physicians and nurses, representing a significant gap in research in other service contexts.

While it is true that health services may have distinct requirements compared to other service domains, the empathic skills cultivated through the identified programs remain relevant in any service context where service employees (SEs) interact with service users (SUs). Our systematic review uncovered four critical skills—communication skills, relationship building, emotional resilience, and counseling—that enhanced SEs' empathic capacity, service quality, and SU satisfaction. Service research has consistently demonstrated that these empathic skills contribute to higher perceived service quality, satisfaction [e.g., 4], and overall service experience [e.g., 5]. This is because, regardless of the nature of the service, interactions between SEs and SUs rely on empathy to establish emotional connectedness [60]. In service encounters, SEs utilize empathic displays as a professional prerequisite to address SUs' needs and facilitate successful service delivery [61]. SEs represent the organization to SUs, fulfill its promises, enhance its reputation and image, and bolster its legitimacy through advocacy. As a result, the service encounter becomes the focal point in SUs' evaluation of the organization [62].

Firstly, communication skills in healthcare settings pertain to the effective exchange of information between parties [e.g., 20, 35], and they are particularly relevant in service interactions that necessitate frequent and delicate exchanges [23, 57]. Similarly, communication skills play a vital role in service organizations by facilitating empathy and addressing the needs of service users (SUs). These skills encompass verbal and non-verbal communication as they convey empathy to others. For example, [63] conducted a study where professional counselors rated the empathic communication of their peers during interactions with clients. The findings revealed that non-verbal bodily cues (such as eye contact, body orientation, trunk lean, and physical distance) accounted for more than twice the variance in ratings compared to verbal messages.

Secondly, relationship-building skills in healthcare settings encompass assisting SEs in establishing effective bonds with SUs through empathy [e.g., 28, 41]. These skills are relevant in healthcare and service settings that involve prolonged interactions with vulnerable SUs [16,

38]. In service organizations, SEs' relationship-building skills are crucial in engaging SUs in reciprocal social interactions, fostering a collaborative relationship known as emotional connectedness [64]. Emotional connectedness relies on SEs' empathic behaviors, such as displaying friendliness [65], actively listening with empathy [4], understanding customers' unique needs through the situational influences of their experiences [66], providing service in a prosocial manner [67], and offering personalized service and advice [68].

Thirdly, in healthcare settings, emotional resilience focuses on equipping SEs with the necessary self-regulation skills to mitigate professional and emotional burnout [e.g., 19, 23, 36]. These skills are particularly relevant in demanding and emotionally exhausting service settings [e.g., 38, 43]. Within service organizations, emotional resilience skills assist SEs in effectively managing the emotional burden associated with empathizing with SUs, which can result in stressful service interactions and emotional distress or burnout [e.g., 69]. Excessive emphasis on sharing SUs' affective states and becoming emotionally entangled with them (referred to as self and other confusion) is more likely to induce emotional distress and burnout instead of fostering prosocial responses in SEs [e.g., 70]. Conversely, individuals who can regulate interpersonal emotions are more likely to experience empathic concern (i.e., a prosocial motivational state that promotes caring and helping; [71], p. 112) toward those in need [72].

Finally, counseling skills in healthcare settings empower SEs to support SUs through strategies such as motivational interviewing (e.g., exploring SU's perspective; 29; 30]. In service organizations, counseling skills help SEs to acknowledge SUs' emotional experiences and to adapt the service delivery accordingly, providing relevant counsel and support to the SUs' situations [e.g., 66, 73].

The findings of our systematic review indicate that empathy training can be easily implemented in the workplace, with no standardized requirement for the number of sessions or duration. The training modules can also take various forms, such as simulation-based training, reflective writing, mindfulness training, or communication training, to align the specific goals of the training with the requirements of the service encounter. Therefore, the empathy training programs identified in the 44 selected studies can be readily implemented and tested in different service settings.

Empathy training programs must be consistent with the type of service users (e.g., customers or patients), their emotions, and the specific service setting. As empathy training can enhance service users' satisfaction or improve SEs' well-being, managers should also determine the intended targets and identify the managerial issues it aims to address. This consideration will help guide the design and implementation of effective empathy training programs in service organizations. This raises several unanswered research questions, providing exciting opportunities for further research. In addition to the limitations discussed in the previous section (refer to Table 4), we have proposed a research agenda in Table 5, highlighting the top priority areas for future research in empathy training for service employees (SEs). This research agenda addresses critical issues and suggests practical approaches for managing existing challenges and opportunities.

### 4.1. How do empathy training programs adapt to the empathy definition?

The meaning of "empathy" training is too broad to be effective, and a clear definition of empathy should be provided to set the training program's goals. Service researchers generally agree that empathy relies on an affective and a cognitive route [74]. Affective (mainly automatic) and cognitive (more controlled) routes are independent but interact to elicit empathic concern (i.e., a motivational state that promotes caring; [71]) and prosocial service behaviors [1]. However, our systematic review showed that empathy training mainly targeted the cognitive route

**Table 5. Research agenda and managerial implications.**

| | KEY RESEARCH QUESTIONS | KEY MANAGERIAL ISSUES |
|---|---|---|
| 1 | How do empathy training programs adapt to the empathy definition? | Design empathy training programs that help SEs to balance affective and cognitive empathy and compassion during service interactions |
| 2 | How do empathy training programs adapt to the SUs' emotions? | Design empathy training programs that help SEs regulate emotions through emotional labor and provide SUs with relevant empathic support. |
| 3 | How do empathy training programs adapt to the service settings? | Design empathy training programs that help SEs to identify and interpret SUs' emotional signals in face-to-face *versus* technology-mediated communication |
| 4 | How do empathy training programs adapt to unconscious biases that affect SEs' empathy for SUs? | Design empathy training programs that help SEs to prevent discrimination during service interactions |
| 5 | How do empathy training programs adapt to the new service triad? | Design empathy training programs that help SEs to integrate service robots during empathic service interactions with SUs |

of empathy, where trainees developed their ability to take others' perspectives and understand them. What about the affective route of empathy? And what about the role of different components, such as empathic concern? For instance, recent studies showed that empathic concern would be more relevant than empathy [75].

Complex social interactions necessitate the simultaneous activation of both affective and cognitive pathways of empathy to gain an accurate understanding of another person's emotional and mental states [76], as well as to evoke empathic concern [75]. Empathic concern—also called compassion—refers to the experience of having genuine feelings for others, which motivates individuals to alleviate their distress through support or consolation [71]. Empathic concern differs from empathy's affective and cognitive pathways, although both processes contribute to its emergence [71]. Empathic concern predicts prosocial behaviors, including sharing, helping, and engaging in mutually beneficial actions [75]. In other words, when interacting with an emotionally affected individual, both affective and cognitive empathy are simultaneously activated, mediating empathic concern [71] and subsequent prosocial behaviors.

Developing a training program where empathy's affective and cognitive routes are unbalanced is risky. Focusing on the cognitive route could foster SEs' social disabilities: deficit in affective sharing while effectively understanding and anticipating SUs' behavioral intentions, taking advantage of them to manipulate them [77]. Focusing on the affective route of empathy (e.g., asking SEs to identify emotionally with SUs) can elicit adverse effects, too. First, it could result in increased emotion regulation to overcome personal distress, which SUs will perceive as a lack of caring. Second, it could lead SEs to respond to SUs' emotions according to rote rules and then make mistakes in judgment [78]. Therefore, future research on empathy training should discuss and test the pros and cons of targeting the affective or the cognitive route of empathy, or empathic concern over empathy, regarding the SEs' professional requirements.

## 4.2. How do empathy training programs adapt to the SUs' emotions?

Our systematic review revealed a significant limitation regarding the lack of attention to service users' (SUs) emotions in empathy training programs. Empathy is a two-way process, and service employees (SEs) must manage the impact of SUs' emotions on their emotional state while dealing with the demands of the service encounter [79]. It is essential to recognize that

empathizing with angry customers in a retail setting differs from empathizing with anxious patients in a healthcare setting. When customers express anger, it can trigger mimetic and aggressive emotional responses in frontline employees (FLEs) through emotional contagion [80]. This poses a challenge as empathizing with angry customers can impair FLEs' ability to display empathy, which is a crucial mediator of prosocial service behaviors [62]. Therefore, empathy training should take into account the emotions of SUs.

For example, consider empathizing with an angry customer in a retail store. When customers express their complaints angrily, it hurts the emotions of SEs. The anger exhibited by customers can elicit mimetic and aggressive responses in SEs through emotional contagion, leading to hostile behaviors that contradict the expected empathic display [79]. Therefore, demonstrating empathy requires SEs to regulate their mimetic response to customer anger through emotion regulation, often called emotional labor [62].

SUs' emotions do not lead to the same empathic support expectation either. Dealing with anxious SUs requires SEs to tap into affective empathy to acknowledge a situation's emotional impact on SUs and provide them with emotional support (i.e., direct anxiety reduction; [80]). Conversely, dealing with angry SUs requires SEs to tap into cognitive empathy to identify and understand the SUs' needs and perspectives, providing them with problem-solving support to alter the situation that elicited anger (i.e., increased cognitive clarity; [80]). Therefore, different SUs' emotions should elicit adaptive SEs' empathic reactions and care. Although it can be challenging to identify one dominant emotion in a specific service context, future research should identify the most recurrent ones and adapt and test the empathy training program accordingly.

## 4.3. How do empathy training programs adapt to the service settings?

Service settings are not discussed in the empathy training programs we reviewed in this study, although they can significantly affect SEs' empathy. Consider the type of service interaction, either face-to-face or mediated by technology. In face-to-face interactions, non-verbal behaviors (e.g., smiling) can "*communicate an empathetic state that facilitates the development of trust and leads directly to cooperative behavior*" [81, p. 10]. However, technology-mediated interactions filter for non-verbal signals of emotion (e.g., facial expressions). It can impair SEs' empathy since sharing and inferring others' emotions depends on unconscious mechanisms of emotion recognition and contagion [82]. Non-verbal cues are also crucial for SEs to convey empathy to customers.

The extent of filtering depends on the features of the medium. For instance, individuals infer others' thoughts and feelings more accurately when they see a full video or hear an audio recording of their interactions, compared to silent videos or transcripts [83]. Therefore, video chat filters fewer signals because it synchronously transmits visual and audio information, whereas more filtering occurs for asynchronous, low-richness media such as email. Moreover, the filtering effect of technology-mediated communication influences affective empathy more than cognitive empathy. For instance, individuals report experiencing more cognitive empathy than affective empathy in text-based interactions [84].

Finally, the length of service interactions should be considered when developing and testing empathy training programs. Interacting with emotional SUs for a few minutes in convenient services like fast-food restaurants does not require the same effort in empathizing as interacting for a few hours or even several days in healthcare settings such as long-term care units.

## 4.4. How do empathy training programs adapt to unconscious biases that affect SEs' empathy for SUs?

Service encounters are social interactions to achieve "a temporary sense of closeness" between SUs and SEs [85, p. 538]. Therefore, SUs and SEs should be matched on their psychological

and personality profiles during service interactions to allow for smoother interactions and greater empathy for each other [3]. However, social group affiliation, such as ethnicity, can impair SEs' empathy [86]. For instance, Joyce Echaquan, an Atikamekw woman who attended healthcare services in North Montreal (Canada) while suffering from pulmonary edema, faced racist slurs from the hospital staff that contributed to her death [87]. In another service context, Starbucks clerks in Philadelphia racially profiled two black customers. They were later arrested based on suspicion of trespassing, although no charges were pressed against them [88]. Those examples illustrate how unconscious biases such as racism can dramatically impair SEs' empathy toward SUs and why they should be addressed in future empathy training programs.

In addition to ethnicity, social closeness affects the ability to empathize and receive empathy. For instance, friends are more accurate at inferring each other's thoughts and feelings in dyadic interactions than strangers [89]. Friends also display increased interactional involvement; they look, smile, and gesture at their partners more often than strangers. Interestingly, even after controlling for this involvement, friends were still better at inferring their partner's mental state. Friends can draw on more events and experiences outside the immediate context when interacting because of their shared knowledge of each other's life. Therefore, interpersonal closeness influences how accurately people infer each other's mental state, and such biases should be addressed in future empathy training programs.

## 4.5. How do empathy training programs adapt to the new service triad?

The shift toward automation of complex processes has significantly influenced service encounters [90], and the COVID-19 pandemic has resulted in a sharp demand increase in service robots (ServBots) to replace SEs [91]. ServBots refer to "*system-based autonomous and adaptable interfaces that interact, communicate, and deliver service to an organization's customers*" [92, p. 909]. ServBots can handle functional operations such as carrying luggage [93] and engage in social interactions with customers through artificial empathy [94]. Therefore, ServBots will be increasingly incorporated into the new service triad—SEs, SUs, and ServBots [95]. In other words, ServBots are more likely to work with SEs rather than replace them. However, it is still unclear how SEs will accept working with ServBots during emotional situations with SUs and how SEs will empathize with SUs while interacting simultaneously with ServBots.

Recent studies show that SEs and ServBots divide tasks according to their nature: ServBots would be responsible for operational tasks, while SEs would be responsible for interactional tasks [96]. However, this new service triad creates a new dynamic at the service encounter. SEs should be trained to provide SUs care and compassion while incorporating the ServBots into the interaction. Future empathy training programs should address this new reality and develop modules where SEs work alongside ServBots during emotional service encounters with SUs.

## 5. Limitations of the current study

Although this review updates current knowledge on existing empathy training programs, it has some limitations. First, given the variability tied to the empathy concept, we may have missed some essential papers. However, our team was composed of experts in the field, and the search strategy was conducted by an information specialist. Second, we didn't include grey literature. It is plausible to believe that, for example, service businesses make their "results" available other than "empirical study" publication. Third, we did not contact the authors of the selected studies to validate our analysis or ask them for more information about, for example, the empathy training programs and corresponding results. Thus, our assessment of the methodological quality is based on what is reported in the articles, and a negative score does not

necessarily mean that the quality is poor but rather that the authors did not report all the information in their publication. Finally, we identified empirical papers only in the domain of health services, although we expected to find articles in service marketing and business. Nonetheless, this absence of results is a result *per se*, as it shows the lack of empirical research on empathy training in the service business and the need to investigate. We believe the research agenda we suggested will foster promising future research.

## Supporting information

**S1 Checklist. PRISMA 2020 checklist.**
(DOCX)

**S1 Appendix. Search strategy.**
(DOCX)

**S1 Table. MMAT.**
(XLSX)

## Author Contributions

**Conceptualization:** Mathieu Lajante, Marzia Del Prete, Beatrice Sasseville, Geneviève Rouleau, Marie-Pierre Gagnon.

**Data curation:** Marzia Del Prete, Beatrice Sasseville, Normand Pelletier.

**Formal analysis:** Mathieu Lajante, Marzia Del Prete, Beatrice Sasseville.

**Funding acquisition:** Mathieu Lajante.

**Methodology:** Mathieu Lajante, Marie-Pierre Gagnon, Normand Pelletier.

**Project administration:** Mathieu Lajante.

**Supervision:** Mathieu Lajante, Geneviève Rouleau, Marie-Pierre Gagnon, Normand Pelletier.

**Validation:** Mathieu Lajante.

**Writing – original draft:** Mathieu Lajante.

**Writing – review & editing:** Mathieu Lajante, Marzia Del Prete, Beatrice Sasseville, Geneviève Rouleau, Marie-Pierre Gagnon, Normand Pelletier.

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
