## [Decision Letter · Decision Letter 0]

4 May 2023

PONE-D-22-28102Empathy training for service employees: A mixed-methods systematic reviewPLOS ONE

Dear Dr. Lajante,

Thank you for submitting your manuscript to PLOS ONE. After careful consideration, we feel that it has merit but does not fully meet PLOS ONE’s publication criteria as it currently stands. Therefore, we invite you to submit a revised version of the manuscript that addresses the points raised during the review process.

We look forward to receiving your revised manuscript.

Kind regards,

Katie Gibson Smith

Academic Editor

PLOS ONE

Journal Requirements:

Reviewers' comments:

Reviewer's Responses to Questions

**Comments to the Author**

1. Is the manuscript technically sound, and do the data support the conclusions?

Reviewer #1: Partly

Reviewer #2: Yes

2. Has the statistical analysis been performed appropriately and rigorously? 

Reviewer #1: N/A

Reviewer #2: N/A

3. Have the authors made all data underlying the findings in their manuscript fully available?

Reviewer #1: Yes

Reviewer #2: Yes

4. Is the manuscript presented in an intelligible fashion and written in standard English?

Reviewer #1: Yes

Reviewer #2: Yes

5. Review Comments to the Author

Reviewer #1: Thank you for the opportunity to review this paper.

My main concerns about the paper are that it is essentially a systematic review of empathy training in the healthcare setting. Despite being framed as a review of papers in the service industry, only papers relating to healthcare were included. I am unsure if this review adds to the many other systematic reviews of empathy training in healthcare published in recent years. The authors present more of a narrative review and there is little discussion on the effectiveness of training (which in my mind would be most relevant). I appreciate the authors comment that this is the first review of empathy training programs in the service research – but as there were no papers outside of the healthcare field identified and included – it is not a review of empathy training for service employees but instead a review of empathy-training in healthcare service.

The authors describe using the TIDieR checklist to report on the details of each individual studies, which is helpful. However, the authors do not comment on whether their systematic review is reported using any guideline, for example PRISMA. This would strengthen the reporting of the review and bring more clarity to why the review was done, what the authors did and what was found.

The authors identify a major limitation of the papers included, that only 50% include a definition of empathy. This is very important and it is good that the authors have included Table 3. The authors however do not give their own definition of ‘empathy training’ in terms of their own inclusion criteria. The inclusion criteria includes ‘at least one training in empathy is identifiable’ but it is unclear how ‘training in empathy’ is determined. As related terms were used in the search strategy (such as emotional intelligence or compassion), were studies included that the authors thought could be considered ‘empathy training’ but were in fact described as something else (e.g. active listening training)? Without a clear definition of what the authors considered to be ‘empathy training’ it would be difficult to say if the papers included were appropriate, particularly as approximately 50% of the papers included did not give a definition of empathy. Further clarity on how papers were included/excluded would be useful – for example were only studies that aimed to foster empathy as a primary outcome included, or were papers that aimed to improve communication skills, with a secondary outcome of fostering empathy also included? More clarity on the outcome variables would also be helpful in understanding how the authors made decisions on which papers should be included/excluded.

Limitations of the selected papers (section 3.9)

The first two sentences report a summary of the findings rather than limitations of the selected papers. It would be helpful for the reader to include a ‘summary of findings’ section and to summarise the findings of the review more clearly here. It would be helpful to include more details summarising each paper in table 2.

The ‘limitations of selected papers’ section does not refer back to the MMAT scores that are discussed earlier in the paper. As the authors went to the trouble of using the tool, it would be helpful to see how the scores they arrived at using the tool link into their assessment of the selected papers limitations. The MMAT scores are described on page 14 under ‘results’ (section 3.1 Description of the included studies and methodological quality assessment) but more clarity on how papers were scored and what scoring involved would be helpful. Perhaps a table in the appendix showing the different MMAT domains and how each paper scored?

Discussion

As the authors point out in the abstract, all 44 included papers focused solely on healthcare (predominantly nursing and physicians) and this significantly limits the generalizability of the findings in other sectors. However, in the discussion section (section 4, page 25-26) the authors suggest the findings can be generalised to other settings “empathy training programs identified…can be easily implemented and tested in other service settings”. This seems quite a leap of faith.

The vast majority of the discussion section is taken up by discussion of the five main research areas the authors believe to be the top priority for future research in empathy training (table 5). There seems to be little connection between the findings of the systematic review (all of which are from a healthcare setting) and the future research priorities discussed. A clearer summary of what the main findings were and a more in-depth discussion of the findings would enhance the paper.

Conclusion

This section includes a discussion on the limitations of the study. I would suggest that a separate heading for ‘limitations of the study’ would be helpful. The authors do report on some limitations, but do not report, for example on the heterogeneity of the papers included in the study. Another limitation of the study is that all the papers come from the same service field (healthcare).

The authors suggest that despite the limitations they describe, their study provides insightful information regarding empathy training programs in service research. It should perhaps be made clear here that the study provides insight to health service research only.

The final section of the conclusion is describing the findings and might be better placed at the beginning of the discussion section.

Reviewer #2: The authors have done invaluable work to gather current evidence on empathy training (mainly health simulation) and to see whether this is similarly applied in a different field (i.e service industries in business studies). Their rational for the need of empathy training in service industries is sound. The literature review approach is carried out thoroughly and transparent.

Their suggestions for further research are an excellent starting point for further research. I dare to say that further fields could benefit from empathy training such as education, leadership and social care (which is partly included in the health simulation).

The areas for some improvements for me are around how the authors explain understand empathy – how it differs from compassion, sympathy and empathic concern. I further would like to see more explanation “on empathy is on a one way process” and social interactions… having said this authors provide an explanation for service encounters (albeit a slightly old reference Siehl et al 1992), yet I would appreciate a few more positions on this (e.g. Luhman, Sacks, Geoffman, Silverman – I’m just noticing that these would also be ‘old’)

Maybe this reference below is of help

Miller, F., & Wallis, J. (2011). Social interaction and the role of empathy in information and knowledge management: A literature review. Journal of Education for Library and Information Science, 122-132.

Are the service encounters always expected to be one to one? How is ‘empathy’ exchange process in a triad or larger group?

All in all I really enjoyed reading this manuscript and would like to thank the authors.

6. PLOS authors have the option to publish the peer review history of their article (what does this mean?). If published, this will include your full peer review and any attached files.

Reviewer #1: No

Reviewer #2: No

---

## [Author Response · Author response to Decision Letter 0]

7 Jun 2023

We addressed all the comments and requests from the reviewers. Our responses are included in the file named "Response to Reviewers."

---

## [Decision Letter · Decision Letter 1]

27 Jul 2023

Empathy training for service employees: A mixed-methods systematic review

PONE-D-22-28102R1

Dear Dr. Lajante,

We’re pleased to inform you that your manuscript has been judged scientifically suitable for publication and will be formally accepted for publication once it meets all outstanding technical requirements.

Kind regards,

Katie Gibson Smith

Academic Editor

PLOS ONE

Additional Editor Comments (optional):

Reviewers' comments:

Reviewer's Responses to Questions

**Comments to the Author**

1. If the authors have adequately addressed your comments raised in a previous round of review and you feel that this manuscript is now acceptable for publication, you may indicate that here to bypass the “Comments to the Author” section, enter your conflict of interest statement in the “Confidential to Editor” section, and submit your "Accept" recommendation.

Reviewer #1: All comments have been addressed

Reviewer #2: All comments have been addressed

2. Is the manuscript technically sound, and do the data support the conclusions?

Reviewer #1: Yes

Reviewer #2: Yes

3. Has the statistical analysis been performed appropriately and rigorously? 

Reviewer #1: N/A

Reviewer #2: N/A

4. Have the authors made all data underlying the findings in their manuscript fully available?

Reviewer #1: Yes

Reviewer #2: Yes

5. Is the manuscript presented in an intelligible fashion and written in standard English?

Reviewer #1: Yes

Reviewer #2: Yes

6. Review Comments to the Author

Reviewer #1: I'd like to thank the authors for their detailed response to my comments and suggestions. I am happy that they have adaquately addressed concerns and have made appropriate changes to their manuscript. I have no further comments or concerns. I enjoyed reading this paper and appreciate the contribution the authors make to the field of empathy research.

Reviewer #2: I'm happy with the changes to the manuscript. well done. I look forward to this manuscript being published.

7. PLOS authors have the option to publish the peer review history of their article (what does this mean?). If published, this will include your full peer review and any attached files.

Reviewer #1: No

Reviewer #2: No

---

## [Editor Report · Acceptance letter]

4 Aug 2023

PONE-D-22-28102R1 

Empathy training for service employees: A mixed-methods systematic review 

Dear Dr. Lajante:

I'm pleased to inform you that your manuscript has been deemed suitable for publication in PLOS ONE. Congratulations! Your manuscript is now with our production department. 

Kind regards, 

on behalf of

Dr. Katie Gibson Smith 

Academic Editor

PLOS ONE